# Improved Diazo-Transfer Reaction for DNA-Encoded Chemistry and Its Potential Application for Macrocyclic DEL-Libraries

**DOI:** 10.3390/molecules26061790

**Published:** 2021-03-22

**Authors:** Selahattin Ede, Mandy Schenk, Donald Bierer, Hilmar Weinmann, Keith Graham

**Affiliations:** 1Bayer AG, Innovation Campus Berlin, 13353 Berlin, Germany; selahattin.ede@nuvisan.com (S.E.); mandy.schenk@nuvisan.com (M.S.); HWeinman@its.jnj.com (H.W.); 2Nuvisan ICB GmbH, 13353 Berlin, Germany; 3Bayer AG, Medicinal Chemistry, 42096 Wuppertal, Germany; donald.bierer1@bayer.com

**Keywords:** DNA-encoded chemical library, DEL, diazo-transfer reaction, macrocycle

## Abstract

DNA-encoded libraries (DEL) are increasingly being used to identify new starting points for medicinal chemistry in drug discovery. Herein, we discuss the development of methods that allow the conversion of both primary amines and anilines, attached to DNA, to their corresponding azides in excellent yields. The scope of these diazo-transfer reactions was investigated, and a proof-of-concept has been devised to allow for the synthesis of macrocycles on DNA.

## 1. Introduction

The power of DNA-encoded libraries (DEL) has become an important and powerful platform to identify new ligands for diseases of pharmacologically interest. Individual DEL libraries can vary significantly in size; however, they typically contain millions of compounds each individually linked covalently to a unique DNA strand. This DNA strand has a unique set of different base pairs that act as a barcode to identify the compound structure that is covalently attached. This complete process has been excellently reviewed and described previously [1,2,3,4,5]. The DEL lead finding strategy has been successfully implied to identify clinical candidates like RIP1 kinase inhibitor GSK3145095 [6] and more recently the autotaxin inhibitor X-165 [7].

The development of DNA-compatible chemical transformations has enabled a larger chemical space to be covered and has been excellently reviewed by Satz et al. [8]. More recent discoveries show that radical chemistry [9], photochemistry [10,11,12], palladium and copper cross coupling reaction [13,14,15,16,17], and reactions carried out in non-aqueous conditions using a solid phase approach are possible [18,19]. Despite these new excellent methods, there is still plenty of need for more methods to further increase the chemical space for DEL. We were interested in developing methods to introduce an azide selectively at a late stage of the library synthesis and use this azide to explore different conjugations, in particularly an intramolecular Huisgen [3 + 2]-cycloaddition to generate medicinal chemistry-like macrocycles.

The diazo-transfer reaction for converting amines to azides has been used previously in organic chemistry (for a brief review see [20]). For more complex molecules, this transformation has been shown to work on solid phase [21], oligodeoxyribonucleotides [22], and proteins [23]. Recently, the work of Gironda-Martínez et al. [24] showed that it was possible to generate azides selectively from the corresponding a-amino group within amino acids conjugated to DNA, via a diazo-transfer reaction, using the relatively safe imidazole-1-sulfonyl azide tetrafluoroborate salt. This method was developed to proceed without copper as a catalyst as the copper could potentially cause DNA-damage. DNA damage should be avoided in DEL-chemistry as this hinders the identification of the DNA-sequence that is the code to identify the ligand attached to the DNA.

## 2. Results and Discussion

### 2.1. Optimization of the Diazo-Transfer Reaction on Two Model Systems

We independently initiated our studies with the DNA-tagged 6-aminohexanoic acid (**1**) as the model substrate for aliphatic amines and DNA-tagged 4-aminobenzoic acid (**2**) for aromatic amines. We decided to use imidazole-1-sulfonyl azide sulfate salt (ISA.H_2_SO_4_) as the diazo donor, as this was commercially available and safe to handle [25]. Our studies used copper(II) sulfate (CuSO_4_) as the catalyst for optimizing the diazo-transfer reaction (Scheme 1), and the results are summarized in Table 1. 

The conversion and integrity of the DNA-conjugated products were analyzed by ultra-high-pressure liquid chromatography−mass spectrometry (UPLC−MS). For optimization studies, we explored different equivalents of ISA.H_2_SO_4_ and CuSO_4_ over different time periods in different basic aqueous solutions. We initially looked at the borate buffer at pH 9.4 with 50 equivalents (eq.) ISA.H_2_SO_4_, and 10 eq. CuSO_4_ at room temperature (RT), and saw a clear improvement in reaction yields with longer reaction times (Entries 1 vs. 2) for the aliphatic (**3**, 81% vs. 100%) and aromatic azide (**4**, 2% vs. 39%) formation. Similar findings were observed for 1 h vs. 16 h using 0.2 M NaHCO_3_ (Entries 4 vs. 5). Thus, we focused on the 16 h time period. Lowering the amount of CuSO_4_ led to full conversion of **3,** but the aromatic azide formation was slow (Entry 6). Heating the reaction (60 °C) with 1 eq. CuSO_4_ gave no noticeable improvement (Entry 7), and detrimental DNA-damage was observed (see Appendix A). Changing to 0.05 M K_2_CO_3_ gave excellent results (Entry 8) for both the aliphatic azide (98%) and the aromatic azide (96%) formation, and this method was chosen to further explore the scope as 1 eq. CuSO_4_, which surprisingly gave a poorer conversion (Entry 9). 

### 2.2. Determining the Scope for the Optimized Diazo Reaction Conditions

At the beginning, we compared our method to a similar method of Gironda-Martínez et al. [24] (using no copper catalyst) and found excellent conversion for different chain lengths (≥95%, Entries 1–6 in Table 2). Only for the a-amino acid (Entry 2), comparable conversion to our protocol was observed and seemed to identify a potential limitation to the method without CuSO_4_ as most of the examples without Cu had the amine activated either in the a-position to a carbonyl or in a benzylic position, whereas with copper no limitations were observed for primary amines. We believe the two methods, with and without copper, are complimentary, potentially allowing for selective diazo-transformations to further expand the chemical space for DEL libraries. As expected, the control reaction with *N*-methylglycine (Entry 7) led to no conversion, highlighting the selectively for this transformation. The scope of the copper-catalyzed reaction was further explored to look at different functional groups, electronic effects, steric hindrance, and heteroaryl amines; the results are summarized in Table 2 (no obvious DNA damaged was visible in these reactions—see Appendix A).

Aliphatic amines with increasing steric hindrance (Entries 8–10) or containing various functional groups (Entries 11–17) also gave excellent conversions (83–100%). Upon further exploration with aromatic amines, we observed excellent conversions for a benzylic amine (Entry 18) and a range of anilines (Entries 19–31) with only the bromo-containing aniline (Entry 28; 63%) and one iodoaniline (Entry 31; 67%) showing slightly poorer conversions. Next, we explored heteroaromatic amines (Entries 32–36) and found excellent conversion for 3-aminopyrazole (Entry 36). With the other analogs, we expected low to no conversion for amino groups ortho to the pyridine or pyrimidine nitrogen, and this was observed for 2-aminopyridine (Entry 34, 20%) and 2-aminopyrimidine (Entry 35, 4%). For those amino group in the meta-position relative to the pyridine nitrogen, we saw excellent conversion for Entry 33 (100%) and a surprisingly lower conversion for Entry 32 (34%).

### 2.3. Off-DNA and on-DNA Macrocycle Formation

The next step was to carry out a proof-of-concept study using our method in producing macrocycles attached to DNA as illustrated in Scheme 2. The first step was to selectively convert an amine (**5**) to its corresponding azide (**6**) and then carry out an intramolecular [3 + 2]-cycloaddition with an alkyne attached to the DNA to give the macrocycle (**7**). The [3 + 2]-cycloaddition has previously been used for synthesizing macrocyclic DEL peptide libraries containing 4–20 natural amino acids [26]. However, we decided to investigate only small macrocycles to contain 3 amino acids and have small ring sizes (n = 11–14). The main drawback we initially anticipated was proving that the diazo-transfer and subsequent cycloaddition were successful since the azide (**6**) and the cyclized adduct (**7**) both have the same mass and potentially have similar retention times in the UPLC-MS. Therefore, we decided to synthesize four peptides (**8**–**11**) as model substrates similar to those that would be synthesized on the DNA and carry out each transformation under the same reaction conditions as surrogate proof that the same transformation would occur when attached to DNA. The peptides were chosen to encompass different features, i.e., aliphatic amines (**8, 9**), aromatic amines (**10**, **11**), ring size, and strain of the macrocycles (*n* = 11, **16**; *n* = 12, **17, 18**, *n* = 13, **19**). We also chose two of the peptides to synthesize with carboxylic acids C-terminal, so that the corresponding azide (**14**) could be coupled to the DNA (step iv) in Scheme 2) and co-injected with the same compound synthesized stepwise on the DNA (Entries 1 and 2 in Table 3).

The peptides **8**–**11** were synthesized using standard Fmoc-solid phase procedures [27]. The diazo-transfer reactions were carried out (Scheme 3) with good-to-excellent yields for azides **12**–**15** (63–98%). The [3 + 2]-cycloadditions were carried out using known DNA-compatible conditions (Cu(OAc)_2_ with ascorbic acid and tris(benzyltriazolylmethyl)amine (TBTA)) [28]. We did not observe the formation of the macrocycle **16** (n = 11); however, the products for the larger ring sizes 12 (**17** and **18**) and 13 (**19**) could be obtained. These findings match a previous work, where cyclizations with smaller ring sizes, e.g., n = 11, were relatively challenging and lower yielding [29].

The next step was to test whether this chemistry could be used to synthesize DEL libraries. Therefore, we decided to synthesize a small simple library (n = 9) to confirm the findings observed for the off-DNA tripeptides. We chose sequences similar to the off-DNA tripeptides (**8**–**11**), i.e., propargylglycine (Pra) as the first amino acid containing an alkyne. The second amino acid was either phenylalanine (Phe), tryptophan (Trp), or β-alanine (β-Ala). The third amino acid was varied to have either glycine (Gly), β-alanine (β-Ala), γ-aminobutyric acid (γ-Abu), 2-aminobenzoic acid (2-Abz), or 3-aminobenzoic acid (3-Abz), and are outlined in Table 3. 

The tripeptides were synthesized on a DNA-headpiece (for structure see the Appendix A) using standard amide coupling and deprotection conditions and were only purified using the cold ethanol precipitation method after each step. The purity for all the tripeptides after the six steps (3 × couple and deprotect) were good-to-excellent as outlined in Table 3, with only Tryptophan showing poorer results as the unprotected indole nitrogen building block was used in the amide coupling steps.

Pleasingly, the diazo-transfer reactions had high conversions, with again only the Tryptophan containing analogs giving poorer results (Entries 4–6). For the intramolecular [3 + 2]-cycloaddition the conversion worked well for most compounds with only the n = 11 ring size formation not being observed, confirmation of the analogous off-DNA reaction (**12** to **16**, Scheme 3). In all these cyclizations,0 we only observed the formation of a dimer with two DNA-headpieces for Entry 1 (see Appendix A).

To confirm our results were valid, we additionally coupled the fluorenylmethyloxycarbonyl (Fmoc) protected analog of tripeptide amine **11** and the tripeptide azides **14** and **15**, then carried out the diazo-transfer reaction and click cyclizations and confirmed the results with co-injection (Entries 1 and 2, see Appendix A for more details).

## 3. Materials and Methods

Comprehensive experimental details can be found within the Appendix A along with the materials and methods. The Appendix A shows the analytical characterization for each reaction and its optimization where applicable.

## 4. Conclusions

In conclusion, we have developed a method for converting both primary aliphatic amines and primary aromatic anilines on-DNA to their corresponding azides in high to excellent yields. The use of an intramolecular alkyne was successfully explored to generate peptidic macrocycles on-DNA. Future work using more medicinal chemistry like scaffolds to generate macrocyclic DEL libraries are currently being investigated.

## Data Availability

Data is contained within the article and Supplementary Material.

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
