# Peer review of "Improved Diazo-Transfer Reaction for DNA-Encoded Chemistry and Its Potential Application for Macrocyclic DEL-Libraries"

_molecules, 2021, doi:10.3390/molecules26061790_

Round 1
Reviewer 1 Report
The authors Selahattin Ede et al in here present an improved Diazo-Transfer Reaction for DNA-Encoded Chemistry and their potential application for macrocyclic DEL-Librar ies. And the conversion of both primary amines and anilines, attached to DNA, to their corresponding azides in excellent yields is poorly understood, which represents a significant knowledge gap. Such knowledge is fundamental to improve DNA synthesis and identify new starting points for medicinal chemistry in drug discovery. The objective of this project is to find a new way to realize the conversion from easy commercial amine group to azide group. Addressing this critical knowledge gap will facilitate the related field goal of synthesis, diagnosis and treatment. So, I highly recommend to publish it after some necessary revision.
- Compared to preciously published papers such as J. Org. Chem. 2004, 69, 2404-2410 which realizing I-group to azid group, the principle and innovation of this paper should be explained more in details.
- In page 3, line 75, the ‘CuSO4’, ‘4’ need to change to Subscript.
- The authors have detected Conversion rate on Table 2. However, they didn’t supply DNA recovered yield. I’d like to know whether the reaction will cause any DNA damage.
- There are many abbreviations in use. Please show the full names before use them, such as ‘eq’.
5. In line 179, the authors need to supply any funding information according to Molecules guide for authors.
Author Response
Dear Sir / Madam
We would like to thank you for taking the time to review our manuscript and for your constructive feedback. Below are the replies to your comments:
Comment 1: This is a good point, but in this publication the primary alcohol on the sugar is converted to the iodo analog and then a SN2 reaction to generate the azide is carried out. The method we established was developed not to touch the DNA backbone which is different from the JOC paper and in line with the previous published method. We did not go into the mechanism as diazo transfer is known in the literature and the mechanism has been proposed (Tetrahedron Lett., 2014, 55, 2917-2920).
Comment 2: The CuSO4 has been changed to the subscript.
Comment 3: In DEL chemistry the recovery yield is not usually included as typicially the recovery is quantitative - I am not sure I have seen this in similar DEL chemistry papers. When we started DEL work we routinely used optical density to measure the concentrations and these always showed quantitative recovery so we stopped measuring.
For the DNA damage we added a few sentences into the text and added data in the supporting information to show the DNA-damage (UPLC and MS).
Comment 4: Good point and we hope we have covered all the abbreviation in the text now.
Comment 5: we received no external funding and wrote the sentence as given in the quidelines in the manuscript.
We would like to again thank the reviewer for their time and comments and we hope the reviewer finds our changes satisfactory and supports this manuscript for publication in Molecules.
Best regards,
Keith Graham - on behalf of all authors
Reviewer 2 Report
In this manuscript, Ede et al. describe a high-yielding and versatile protocol to convert DNA-appended primary amines into azides. The procedure expands a protocol previously reported by Gironda-Martinez et al.. Advances of this paper are the inclusion of CuSO4 to enhance reaction yields and the identification of an optimized reaction protocol. The conversion of amines to azides is of high interest to scientists synthesizing DNA-encoded libraries and therefore this manuscript is important. Two major aspects need to be addressed before publication:
First, the selling point of the paper is that the addition of CuSO4 enhances the reaction yield over protocols without the metal catalyst. The authors compare the two reactions in entries 1-6 (table 2) and several of the tested amines provide low yields for the protocol without Cu(II) illustrating the need for the metal. However, there is a discrepancy between this conclusion and the results presented in the paper by Gironda-Martinez, who reported high yields for a considerable scope of amines. The amines used for comparison in this paper (3-6) have not been tested by Gironda-Martinez and all are aliphatic amines. Amines that provided high-yielding conversions in the Gironda-Martinez paper, like for example benzylamine, have not been tested with the metal-free protocol. Additional direct comparisons are needed to make conclusions about the need for Cu(II) and provide insight about which types of amines require Cu(II) (for example, there is an clear difference in yields for aromatic amines, and it is interesting to notice that most examples in the Gironda-Martinez paper are amines with a C=O bond at the α/β-position).
Second, the authors claim that the Cu(II) does not damage the DNA. However, this assertion is not rigorously supported by evidence. Additional information on experimental procedures and results are needed to make such a conclusion.
Minor points:
Line 22: "...is essentially millions of compounds..." DNA-encoded libraries come in all sizes and this statement is misleading as written.
Line 37: There are a couple of publications on conversion of amines to azides predating the Gironda-Martinez paper; although these do not involve DNA-encoded libraries, they should still be cited.
Line 38: convert "alpha-amino" to "α-amino"
Throughout the text the authors use a "." for chemical formulas or "*" in the supporting information. These should be replaced by "⋅".
Line 56: "...saw a clear time-dependent improvement..." is unclear. Suggestion: "...saw a clear improvement in reaction yields with longer reaction times..."
Line 61: "detrimental DNA-damage was observed". How was this observed? The data should be shown in the SI.
Line 68: "...further explore the scope..." The scope of what?
Line 73: "Gironda-Martinez et al.." Remove extra point.
Table 2: Many of the structures are cut off. Make sure that all structures are fully visible.
Scheme 1: The caption mentions a green circle. In my copy there are only blue and a grey but no green circles.
Line 146: "... on a DNA-headpiece..." Refer to SI for structure of headpiece.
Line 157: ")" is missing.
Supporting information: The authors should carefully read over the SI to make sure that all compound names are correct and fix other mistakes. For example, on page 11, 6-aminobenzoic acid is misspelled and it should be 4-aminobenzoic acid; on page 12 it the structure name is given in German.
Author Response
Dear Sir / Madam
We would like to thank the reviewer for taking the time to review our manuscript and to give constructive feedback. Our replies and comments to your comments are below:
First, the selling point of the paper is that the addition of CuSO4 enhances the reaction yield over protocols without the metal catalyst. The authors compare the two reactions in entries 1-6 (table 2) and several of the tested amines provide low yields for the protocol without Cu(II) illustrating the need for the metal. However, there is a discrepancy between this conclusion and the results presented in the paper by Gironda-Martinez, who reported high yields for a considerable scope of amines. The amines used for comparison in this paper (3-6) have not been tested by Gironda-Martinez and all are aliphatic amines. Amines that provided high-yielding conversions in the Gironda-Martinez paper, like for example benzylamine, have not been tested with the metal-free protocol. Additional direct comparisons are needed to make conclusions about the need for Cu(II) and provide insight about which types of amines require Cu(II) (for example, there is an clear difference in yields for aromatic amines, and it is interesting to notice that most examples in the Gironda-Martinez paper are amines with a C=O bond at the α/β-position).
Our reply: This a very good point and we didn't clarify this enough in our manuscript. There was a benzylic-like amine in the Gironda-Martinez paper (Figure 1, Entry 12) and ours (Table 2 entry 18). We have changed the text in the manuscript which we hope makes this clearer. The suggestion to do the benzylic amine without copper is an excellent one and I think the yield would be very good as this demonstrated by Gironda-Martinez paper (Figure 1, Entry 12). Unfortunately, to carry out additional experiments will not be possible as Bayer sold the a majority of Berlin-based research resources to Nuvisan last year and none of the authors remain at Bayer in Berlin. I personally started a new position with Boehringer-Ingelheim and have no facilities to do these useful experiments. We hope the reviewer agrees to our additions, without extra experiments.
Second, the authors claim that the Cu(II) does not damage the DNA. However, this assertion is not rigorously supported by evidence. Additional information on experimental procedures and results are needed to make such a conclusion.
Our reply: We thank the reviewer for pointing this out, we have added the UPLC chromatograms and MS data to show the clear DNA degradation to the supporting information (pages 11-12) and have changed or added some wording in the text to address this. We hope the reviewer finds these changes satisfactory.
To the minor points:
Minor points:
Line 22: "...is essentially millions of compounds..." DNA-encoded libraries come in all sizes and this statement is misleading as written.
Our reply - reworded and hopefully ok now.
Line 37: There are a couple of publications on conversion of amines to azides predating the Gironda-Martinez paper; although these do not involve DNA-encoded libraries, they should still be cited.
Our reply: This is a good point and we added a sentence to the text and included references.
Line 38: convert "alpha-amino" to "α-amino"
Our reply: has been changed
Throughout the text the authors use a "." for chemical formulas or "*" in the supporting information. These should be replaced by "⋅".
Our reply: thank you for spotting this and we have changed it
Line 56: "...saw a clear time-dependent improvement..." is unclear. Suggestion: "...saw a clear improvement in reaction yields with longer reaction times..."
Our reply: thank you for the sentence, we change taken this into the revision.
Line 61: "detrimental DNA-damage was observed". How was this observed? The data should be shown in the SI.
Our reply: another good point, we have added data into the SI (pg11-12) to support this statement
Line 68: "...further explore the scope..." The scope of what? –
Our reply: we have reworded this sentence and added more explanation.
Line 73: "Gironda-Martinez et al.." Remove extra point.
Our reply: has been changed
Table 2: Many of the structures are cut off. Make sure that all structures are fully visible.
Our reply: We didn't see this in the word document and am not sure where this happens. We have reformated the Table and hope that everything is clearly visible now. We have asked the editor for support here if this is still not clear.
Scheme 1: The caption mentions a green circle. In my copy there are only blue and a grey but no green circles.
Our reply: I am actually colourblind so I hope everything is now with the right colour - thank you for noticing this.
Line 146: "... on a DNA-headpiece..." Refer to SI for structure of headpiece. Statement added
Our reply: Structure of the headpiece has been added to the SI.
Line 157: ")" is missing.
Our reply: has been changed - thanks for spotting this.
We would again like to thank the reviewer for their comments as they have improved the manuscript. We hope the reviewer finds our replies and changes satisfactory and that our manuscript would be suitable for publishung in Molecules.
Best regards,
Keith Graham
Round 2
Reviewer 2 Report
The authors have largely addressed my comments. I would have loved some additional direct comparisons between the present method and the previously reported results; however, given the authors' situation, I have no issues with publishing it as it is.